# Maternal Pre-Pregnancy Cardiovascular Risk Factors and Offspring and Grandoffspring Health: Bogalusa Daughters

**DOI:** 10.3390/ijerph16010015

**Published:** 2018-12-21

**Authors:** Emily W. Harville, John W. Apolzan, Lydia A. Bazzano

**Affiliations:** 1Department of Epidemiology, Tulane University School of Public Health and Tropical Medicine, New Orleans, LA 70112, USA; lbazzano@tulane.edu; 2Pennington Biomedical Research Center, Louisiana State University System, Baton Rouge, LA 70808, USA; john.apolzan@pbrc.edu

**Keywords:** body mass index, skinfold thickness, lipids, blood, birth weight, gestational age, transgenerational

## Abstract

Both maternal pre-pregnancy body mass index (BMI) and gestational weight gain have been associated with cardiovascular health in the offspring beyond two generations. A total of 274 daughters (aged 12–54) of 208 mothers who participated in the Bogalusa Heart Study were interviewed about their reproductive history. Mothers’ data was taken from the original study, and cardiovascular measures at the visit prior to pregnancy were correlated with daughter’s measures. Maternal pre-pregnancy BMI, skinfold, and waist circumference were examined as a predictor of daughters’ blood pressure, lipids, and glucose, as well as a predictor of birthweight and gestational age of grandchildren. Maternal pre-pregnancy BMI was associated with higher blood pressure and lower low-density lipoprotein (LDL) and cholesterol in the daughters. Most maternal cardiometabolic risk factors were not associated with grandchildren’s birth outcomes, even though higher cholesterol and LDL was associated with lower gestational age, and higher BMI and skinfold thickness with an increased risk of preterm birth. In this pilot study, some associations were found between maternal adiposity and cardiovascular risk, daughters’ cardiovascular risk, and grandchild birth outcomes. Lack of conclusive associations could be due to a true lack of effect, effects being primarily mediated through daughter’s BMI, or the low power of the study.

## 1. Introduction

The fact that the groups that have experienced the most oppression and discrimination across generations often have the worst health outcomes, and racial disparities persist across populations with comparable access to health care and, in otherwise low-risk groups [1,2,3,4], suggests that health is affected by factors beyond an individual’s immediate circumstances. The developmental origin of health and disease theory indicates that adult health can be programmed by in utero exposures [5].

Both maternal pre-pregnancy body mass index (BMI) and gestational weight gain have been associated with cardiovascular health in the offspring. High pre-pregnancy BMI predicts adiposity, insulin resistance, lipids, and blood pressure in childhood [6,7,8]. The Jerusalem Perinatal Study also found that greater maternal pre-pregnancy BMI, independent of gestational weight gain and confounders, was significantly associated with higher adult offspring BMI, waist circumference, systolic and diastolic blood pressure (SBP and DBP), insulin, and triglycerides [9]. However, these studies generally find that the majority of the effect is due to offspring BMI [8,9]. Therefore, it may ultimately not prove much beyond the fact that obesity runs in families, which is a fact well-known to be due to both genetic and environmental factors [10]. However, few studies have examined other indicators of maternal adiposity, such as weight circumference.

Some recent work has been extended beyond two generations. In the Panel Study of Income Dynamics, grandparental obesity was found to be associated with grandchildren’s obesity, independent of parental BMI [11]. Prenatal exposure to famine was associated with hyperglycemia in generations 2 and 3 in a Chinese study [12]. The Overkalix (Sweden) study of historical harvest records [13] for 317 people found that, if the paternal grandmother experienced changes in food supply year to year in early life, the granddaughter had increased risk of cardiovascular mortality. Such associations were not found with the experience of the maternal grandmother or either maternal or paternal grandfather.

The mirroring of disparities in birth outcomes and cardiometabolic health [14,15,16] also suggests that it is worthwhile to examine the interrelationship between cardiovascular and reproductive health. A higher incidence of cardiovascular mortality was associated with lower grandchild birthweight in a very large study in Norway [17]. For diabetes mortality, U-shaped associations with maternal grandmother’s diabetes mortality risk were seen for grandchild birthweight, while, for all other grandparents, higher diabetes mortality was associated with lower birthweight [17]. However, a study in Maltese women [18] found that, while generation 1 BMI was directly linked to generation 2 birthweight and BMI, there was no association with generation 3 birthweight. Metabolic syndrome in the generation 1 also did not produce any changes in the birthweight of generation 2 or 3. Our previous analysis in the existing Bogalusa Heart Study (BHS) data [19] found that pre-pregnancy cardiometabolic risk factors (BMI, triglycerides, low-density lipoprotein (LDL), glucose) were associated with birthweight and gestational age in generation 3.

Because of the difficulty of performing multigenerational prospective studies, many previous studies are Scandinavian and take advantage of those countries’ ability to link across databases but provide limited racial and socioeconomic diversity. Our previous BHS analysis did not purposefully recruit second and third generation participants, relying instead on a retrospective linkage. In this study, we intentionally recruited the daughters of former BHS participants, with the goal both of testing feasibility for this type of study design, and examining maternal pre-pregnancy cardiovascular health and daughters’ cardiometabolic risk factors, and grandchild birth outcomes, in the second and third generations of the Bogalusa Heart Study. We hypothesized that women with worse cardiovascular health prior to pregnancy would have daughters with cardiovascular health, and grandchildren with lower birthweight and gestational age, even controlling for generation 2 BMI.

## 2. Materials and Methods

This study included 274 daughters of 208 female Bogalusa Heart Study participants. The Bogalusa Heart Study (BHS) is a series of studies of cardiovascular health in a semirural, biracial population (65% white and 35% black), founded by Dr. Gerald Berenson in 1973, which generated some of the seminal literature relating childhood risk factors to adult cardiometabolic health [20,21]. Furthermore, 1803 women participated in the Bogalusa Babies study, which was a study of reproductive health and pregnancy outcomes nested in BHS. Women were interviewed by phone or in the clinic about their pregnancy history. They had an estimated 1653 daughters in the relevant age range (12+). The Bogalusa Daughters project gathered data on female offspring ages 12 and older of participants in the Bogalusa Babies study. (Age 12 was chosen as the average age of menarche for girls as well as an age when participants were likely to be able to understand the study topic and thus give informed assent). Compared with the overall parous Babies study population, the included group was more likely to be black, more likely to have smoked, and had less education [22] (Harville et al., under review). This study was approved by the Institutional Review Board of Tulane University (#862715).

Mothers’ data was taken from the original study, and cardiovascular measures at the visit prior to pregnancy were identified. For the daughters, study procedure varied slightly depending on the age and geographic location of the participants. Bogalusa Babies participants (mothers) were contacted by mail and phone, and contact information for their daughters was obtained. The daughter was then contacted by telephone. Socio-demographics were updated. If she was able to attend a clinic visit, then blood pressure, height, and weight were measured and a fasting blood draw was taken (details of measures below). An interview was conducted about her reproductive history, and medical records were requested about her pregnancies. If the daughter was aged 12–17, a modified version of this protocol was implemented, with fewer questions about pregnancy. Daughters who lived too far away to visit the clinic, or who did not wish to, were interviewed by telephone. If they were willing to provide biological measures but did not live near Bogalusa, they were invited to visit a local LabCorp for lipid and glucose measurements.

### 2.1. Cardiovascular Measures

Measures were taken with a similar protocol in both generations even though, in a few cases, the protocol in early years (mothers) was different than that for the daughters’ follow-up. Anthropometric measurements were conducted on individuals in light clothing without shoes using a standard protocol. Determinations of height (to nearest 0.1 cm) and weight (to nearest 0.1 kg) were performed in duplicate. Triceps and subscapular skinfold thickness and waist and hip circumferences were measured in triplicate (mothers only, waist circumference was not measured in early years of BHS). For mothers, blood pressure was measured in duplicate by trained research technicians. For daughters, blood pressure was measured using the HEM 907 ZL Non-Invasive Blood Pressure Monitor by Omron HealthCare CO, LTD, Kyoto, Japan. This instrument provides BP and heart rate, and a total of four readings are taken and are averaged for analysis. Plasma glucose was measured using the hexokinase method. Serum total cholesterol and triglycerides were measured by enzymatic procedures. LDL cholesterol was estimated by using the Friedwald equation.

### 2.2. Reproductive Measures

Adult women were interviewed about their age at menarche, gravidity, parity, fertility issues, birthweight, and gestational age of their children using a standardized form. These questions have been used in the Bogalusa Babies study and self-report for pregnancy history is generally reliable [23,24,25]. Adolescents (<18) were asked about age at menarche and any children they had. If they reported having children, information about those pregnancies was gathered. Low birthweight (LBW) was defined as birthweight < 2500 g, and preterm birth (PTB) was defined as birth three or more weeks early. If a woman said that her child had been born on time but did not specify further, gestational age at birth was imputed as 40 weeks.

### 2.3. Analysis

Descriptive data were calculated using frequency tabulations and measures of central tendency and variability. Racial differences in cardiovascular and reproductive outcomes were examined by using t-tests for continuous measures and chi-square tests for categorical variables. Cardiovascular measures were compared across generations. If the mother had more than one BHS visit, the study visit that occurred prior to the pregnancy with the daughter was determined. Given the study design, this did not occur at a standardized time before pregnancy. Median time between this visit and daughter’s birth was four years. Cardiovascular measures at the visit prior to pregnancy were compared to the daughter’s measures by examining correlations between daughter’s risk factors and her mother’s pre-pregnancy and mean adult measures.

Cardiometabolic predictors were standardized, so that all effect estimates are presented for a 1-SD increase. Maternal pre-pregnancy adiposity, as indicated by BMI, skinfold, and waist circumference at the study visit prior to pregnancy, was examined as a predictor of daughters’ cardiovascular risk. Maternal pregnancy weight gain during the pregnancy with the daughter was also examined as a predictor of daughters’ cardiometabolic risk. Generalized estimating equations (linear models), with control for mother’s age, daughter’s age, daughter’s BMI, and race were used to predict daughters’ cardiovascular risk factors with control for clustering within family. For other measures of adiposity, a final model also controlled for maternal BMI. Missing data were not extensive (see Table 1), but multiple imputation (proc mi and mianalyze) were used to control for missing data on confounders. All models with 200 or more observations were examined for interaction with race, using interaction terms and stratification.

Maternal pre-pregnancy cardiovascular risk was also examined as a predictor of the birthweight and gestational age of the first grandchild as well any LBW or PTB across all pregnancies. Again, generalized estimating equations (linear and logistic models) with control for maternal and daughter’s age and daughter’s BMI were used to control for clustering within family.

## 3. Results

Fifty-five percent of participants were black and 45% were white (Table 1), and slightly more than half had at least one pregnancy. Daughters were on average 27 years old at the time of the interview, with a range of 12–54 years old, and mean age during the first pregnancy was 20.6 years. Black women had a higher mean age at interview (28.1 vs. 25.8, *p* = 0.02), a lower mean age at first pregnancy (19.7 vs. 22.5, *p* < 0.01), a higher mean BMI (33.3 vs. 29.9, *p* < 0.01), a higher mean blood pressure (*p* < 0.05, data not shown), a lower mean and median triglycerides (*p* < 0.01), and lower birthweight at first pregnancy (3081 vs. 3289 g, *p* = 0.03). There were no racial differences in LDL, high-density lipoprotein (HDL), or glucose (data not shown).

Mothers’ and daughters’ risk factors were correlated at between 0.2 and 0.3. When limited to measures taken prior to pregnancy on mothers, correlations were not as strong (Table 2).

Maternal pre-pregnancy BMI was associated with higher blood pressure, lower LDL, and lower total cholesterol levels in the daughters (Table 3). Maternal subscapular skinfold was associated with higher diastolic blood pressure, but this result disappeared when BMI was adjusted for, while total cholesterol was higher only after BMI was adjusted for. Waist circumference was associated with lower LDL and total cholesterol, but those associations disappeared after BMI was adjusted for. Similarly, the association between higher gestational weight gain and higher triglycerides disappeared after adjustment for confounders. No interactions with race were statistically significant, but the association between maternal BMI and blood pressure appeared to be stronger in black women (adjusted beta for SBP in black women 3.16, Standard Error (SE(b)) 1.21, *p* = 0.01, for white women, −0.54 (0.92), *p* = 0.55, for DBP, 2.27 (1.05), *p* = 0.03, 0.54 (0.93), 0.56). The association between maternal BMI and HDL was stronger for white women (adjusted beta in white women 7.41 (1.55), *p* < 0.01, for black women, −0.81 (1.48), *p* = 0.58). The association between pregnancy weight gain and triglycerides appeared stronger in black women (adjusted beta 7.03 (4.15), *p* = 0.09) than in white women (−5.11 (4.48), *p* = 0.25).

Maternal cardiometabolic risk factors were associated with birth outcomes in grandchildren only to a limited extent (Table 4). DBP was associated with a higher risk of a LBW baby (attenuated after adjustment for confounders). Higher cholesterol and LDL was associated with lower gestational age, and higher BMI and skinfold with an increased risk of PTB. There was also a tendency for triglycerides to be associated with lower birthweight.

## 4. Discussion

This study examined possible prenatal and multigenerational effects of cardiometabolic risk factors on offspring cardiovascular health and grand-offspring birth outcomes. In a few cases, we found that maternal pre-pregnancy adiposity predicted daughters’ adolescent or adult cardiovascular risk, even controlling for her own BMI. Maternal pre-pregnancy BMI was associated with higher blood pressure, lower LDL, and lower total cholesterol levels in the daughters, and the maternal subscapular skinfold was associated with higher daughters’ total cholesterol. In addition, maternal BMI and blood pressure was stronger in black women, while the association between maternal BMI and HDL was stronger for white women. A previous small study comparing offspring of overweight and normal-weight women [6] found a significant difference in blood pressure, which is similar in size to that seen in black women in our study (Some other studies use a different scale of exposure: comparing obese and non-obese participants, or per one-unit change rather than one-SD. For purposes of these comparisons, the effect sizes for these studies were approximated by converting to a 1-SD effect based on their published SDs). They also found large differences in metabolic factors (though not glucose, the only similar factor we measured directly). Another large study found the offspring of obese women to have higher blood pressure compared to those in other BMI categories, but no differences in cholesterol, and small to no differences in triglycerides [7]. This study also found differences in blood pressure, but not as large as we did. A previous analysis of the Jerusalem Perinatal Study found very similar effect sizes as we did for the effect of maternal pre-pregnancy BMI on systolic and diastolic blood pressure, a similar lack of effect on glucose, and larger effect sizes for HDL and LDL [9].

In addition, there were some associations between maternal pre-pregnancy cardiovascular risk and grand-offspring gestational age even though more associations tested were null. Diastolic blood pressure was associated with a higher risk of LBW. Higher cholesterol and LDL are associated with lower gestational age. Higher BMI and skinfold are associated with an increased risk of PTB and triglycerides are associated with lower birthweight. Our previous study of a different sample within BHS found stronger effects for glucose on birthweight and similar effect sizes for lipids [19]. We are not aware of other studies addressing this precise topic.

Effects of maternal and grandmaternal adiposity on offspring health are biologically plausible. Several animal studies have addressed transgenerational relationships among birth outcomes, adiposity, and glucose/insulin metabolism. In Wistar rats, low protein diet was associated with low birthweight and then increased glycemia in generation 2 and higher birthweight in generation 3 [26]. Trunk diameter at birth and back and intramuscular fat were larger in grand-offspring of overfed and underfed swine. Lipid levels were higher from the overfed group [27]. Generation 3 lambs had elevated glucose levels after generation 1 overfeeding. Total fat at birth was higher and thoracic girth lower even though there was no difference in birthweight [28]. In addition, a restricted diet during pregnancy or lactation led to offspring with evidence of insulin resistance in generations 2 and 3 in Wistar rats [29], and uteroplacental insufficiency leading to low birthweight in generation 2 was associated with changes in insulin response in generation 3 Wistar-Kyoto rats [30]. In female Sprague-Dawley rats, a high-fat diet during pregnancy and lactation was associated with increased body weight and glucose intolerance in generation 3 offspring due to changes in β-cell function and proliferation [31].

BMI is a general indicator of overall adiposity, but, in some cases, waist circumference or skinfold may be a stronger predictor of adverse health [32,33]. For such factors, adjustment for BMI may indicate visceral or subcutaneous fat. Excess abdominal fat is hypothesized to affect health via fat storage and metabolic dysfunction around internal organs and induction of worsened insulin resistance, lipoprotein metabolism, and blood pressure [34]. Overall, we found higher maternal BMI to be associated with higher blood pressure. Contrary to the hypothesis, however, total cholesterol and LDL were lower. For skinfold, adjustment for BMI created a positive association, with higher skinfold predicting higher cholesterol for a given BMI. The reasons for this are difficult to determine, given that abdominal fat is generally more strongly associated with cardiometabolic health than subcutaneous fat [35,36]. One previous study did note a stronger association with maternal glucose during pregnancy for subcutaneous fat than waist circumference [37], which means it is possible that pregnancy-related adiposity functions differently.

Epigenetic research has provided a possible mechanism by which social factors might get “under the skin” to impact health and engender health disparities at the population level [38]. Epigenetic variation contributes to variation in gene expression and risk for cardiovascular disease [39]. Changes in methylation associated with poor health could occur following social adversity at any point in the life course [40], and can even be transferred to the next generation [41]. This could help explain persistent health disparities observed even in those whose adult life circumstances are not obviously deprived [42]. When we examined racial differences in the associations, we did find some to be stronger in black women and some in white, which suggests that future studies with sufficient power to address this in detail should be conducted. Epigenetic differences associated with pre-pregnancy maternal BMI have been found in offspring, and could indicate a mechanism by which generation 1 effects could be translated to generation 3 [43].

This study was a pilot work, partially to test feasibility of two-generation recruitment, and to provide initial data on effect sizes. As such, many comparisons were underpowered, and control for confounding was limited. This is especially the case for associations where the exposure was measured only in a subgroup (such as waist circumference), or for whom not all women were eligible (birth outcomes were limited to those who had given birth). Future studies should aim for larger sample sizes that will address these limitations and allow for robust control for sociodemographic, environmental, and genetic confounders and clustering within family. That said, our results and effect sizes were generally consistent with previous studies.

An additional limitation is the lack of information on the fathers and grandfathers of the study participants. Future studies should gather information on both parents and grandparents, if possible, to allow for control for lifestyle factors and for testing epigenetic hypotheses (which may arise from either parent, and in some studies, have indicated stronger paternal effects [13,44]).

## 5. Conclusions

Future studies should include larger samples, paternal measures, mechanistic indicators, and control for confounding that allows for robust conclusions on this important topic. Our results were similar in effect sizes and outcomes to other multigenerational studies. In summary, we found relationships between some measures of maternal adiposity, daughters’ cardiovascular risk factors, and grandchild birth outcomes.

## Figures and Tables

**Table 1 ijerph-16-00015-t001:** Descriptive data, pilot study of daughters of Bogalusa Heart Study participants.

Descriptor	Daughters (*N* = 274)						Mothers (*N* = 208)				
	*N*	%					*N*	%			
Race						Race					
white	123	45.1				white	103	49.5			
black	150	55.0				black	105	50.5			
Ever smoked						Ever smoked					
ever	63	23.0				ever	96	46.2			
never	211	77.0				never	91	43.8			
						not during pregnancy, no other info	21	10.1			
Education						Education					
<18 years	41	15.0				<18 years					
<high school	32	11.7				<high school	33	15.9			
high school	78	28.5				high school	86	41.4			
associate’s/some college	81	29.6				associate’s/some college	55	26.4			
College+	42	15.3				College+	34	16.4			
Number of pregnancies (18+ years)						Number of pregnancies (18+ years)					
0	120	43.8				0					
1	56	20.4				1	13	6.3			
2+	98	35.8				2+	195	93.2			
	mean	median	SD	min	max		mean	median	SD	min	max
current age (*N*= 274)	27.0	27.0	8.4	12	53.2	current age (*N* = 198)	46.9	47.0	6.8	28.0	61.0
age at 1st pregnancy (*N*= 137)	20.7	20.0	4.3	13.0	32.7	age at 1st pregnancy (*N* = 198)	20.5	19.8	4.2	14.3	33.7
age at menarche (*N* = 271)	12.6	12.0	1.8	8.0	20.0	age at menarche (*N* = 98)	12.6	13.0	1.6	9.0	18.0
birthweight of first baby (*N* = 132)	3160	3175	527	907	4323	birthweight of first baby (*N* = 196)	3183	3175	646	907	5613
						age at pre-pregnancy visit (*N* = 189)	18.7	17.2	6.0	5.6	35

SD: Standard Error; min: minimum; max: maximum.

**Table 2 ijerph-16-00015-t002:** Adiposity and cardiometabolic health across generations. The Bogalusa Daughters study.

Cardiovascular Risk Factor	Daughters (*n* = 222)						Mothers (*n* = 198)					
							pre-pregnancy					
	mean	median	SD	min	max		mean	median	SD	min	max	r * (*n* = 161)
BMI *	31.7	30.4	9.1	14.1	71.1		23	21.3	5.6	13.4	49.2	0.11
systolic blood pressure (mmHg)	118.3	116	13.3	84	175		107	107	9.4	83.3	145	0.11
diastolic blood pressure (mmHg)	75.3	75	11	51	112		68.1	68	7.4	45.7	87.3	0.04
HDL (mg/dL)	51.6	49.5	13.8	21	99		168.5	167.5	30.6	100	295	0.01
LDL (mg/dL)	99.9	97	29.4	37	228		57.3	56	16.9	4.4	108.6	0.07
total cholesterol (mg/dL)	170.1	168	34.7	18	289		100.1	97.3	27.2	51.3	192.8	0.18
triglycerides (mg/dL)	93.2	77	50.3	30	376		78.6	68	45.4	23	417	0.25
glucose (mg/dL)	97.5	93	26.6	66	323		81.5	80	8.4	62	107	0.21
	mothers											
	adult											
	mean	median	SD	min	max	r (*n* = 170)						
BMI *	28	26.2	6.7	17.9	46.8	0.37						
systolic blood pressure (mmHg)	110.9	109.7	10.1	92.7	147.1	0.17						
diastolic blood pressure (mmHg)	73.6	72.9	6.8	56.3	96.9	0.23						
HDL (mg/dL)	52.9	53	11.4	29	99	0.31						
LDL (mg/dL)	114.1	14.8	25.3	60.9	187.8	0.37						
total cholesterol (mg/dL)	181.3	182.4	27.8	126.5	260	0.36						
triglycerides (mg/dL)	97.6	89	55.6	33	548.3	0.33						
glucose (mg/dL)	84.7	82	13.6	65	171.3	0.19						

* BMI, body mass index. HDL, high-density lipoprotein. LDL, low-density lipoprotein; *p* < 0.05 for all r > 0.1, *p* > 0.05 for all r < 0.1.

**Table 3 ijerph-16-00015-t003:** Maternal adiposity as a predictor of child cardiovascular risk, the Bogalusa Daughters Study.

Risk Factor	BMI (*n* = 200)									Subscapular Skinfold (*n* = 200)								
	Unadjusted			Adjusted *						Unadjusted			Adjusted *			Adjusted + BMI		
	beta	SE	*p*	beta	SE	*p*				beta	SE	*p*	beta	SE	*p*	beta	SE	*p*
systolic blood pressure (mmHg)	0.13	0.84	0.87	1.77	0.85	0.04				0.40	0.83	0.63	1.38	0.87	0.11	0.00	1.31	1.00
diastolic blood pressure (mmHg)	0.42	0.86	0.62	1.69	0.78	0.03				1.13	0.68	0.10	1.42	0.56	0.01	0.30	1.08	0.78
HDL (mg/dL)	−1.19	0.95	0.21	−1.09	0.96	0.26				−1.30	0.91	0.16	−0.83	0.92	0.37	0.50	1.35	0.97
LDL (mg/dL)	−4.09	2.23	0.07	−3.93	1.86	0.03				−0.68	2.10	0.75	−1.12	1.86	0.55	5.19	3.00	0.08
total cholesterol (mg/dL)	−5.11	2.36	0.03	−5.29	2.14	0.01				−1.10	2.33	0.64	−1.77	2.17	0.42	6.33	3.24	0.05
triglycerides (mg/dL)	1.55	3.20	0.63	0.11	3.72	0.98				5.07	2.86	0.08	2.63	3.11	0.40	3.87	4.91	0.16
glucose (mg/dL)	−0.75	1.96	0.70	−0.78	1.95	0.69				1.70	1.65	0.30	1.36	1.44	0.34	5.30	3.63	0.14
	gestational weight gain (*n* = 219)									waist circumference (*n* = 101)								
	unadjusted			adjusted *			adjusted + BMI			unadjusted			adjusted *			adjusted + BMI		
	beta	SE	*p*	beta	SE	*p*	beta	SE	*p*	beta	SE	*p*	beta	SE	*p*	beta	SE	*p*
systolic blood pressure (mmHg)	−1.11	0.83	0.18	0.63	0.86	0.46	0.82	0.85	0.34	−0.94	0.93	0.31	−0.25	0.91	0.78	−0.76	2.57	0.77
diastolic blood pressure (mmHg)	−0.94	0.77	0.23	0.16	0.71	0.82	0.11	0.67	0.87	0.34	0.86	0.69	0.20	0.79	0.80	−0.20	1.95	0.92
HDL (mg/dL)	−0.61	0.76	0.43	0.00	0.65	1.00	−0.23	0.64	0.72	−2.44	1.22	0.05	−1.38	1.36	0.31	−7.04	4.48	0.12
LDL (mg/dL)	0.21	2.07	0.92	0.45	2.05	0.83	−0.21	2.13	0.92	−3.67	3.26	0.26	−4.39	2.35	0.06	4.75	7.59	0.53
total cholesterol (mg/dL)	1.39	2.27	0.54	0.73	2.26	0.75	−0.22	2.32	0.93	−6.10	3.47	0.08	−7.68	2.89	0.01	0.44	7.24	0.95
triglycerides (mg/dL)	7.28	3.10	0.02	1.48	3.06	0.63	0.68	3.16	0.83	1.02	3.67	0.78	−5.32	3.79	0.16	15.51	11.61	0.18
glucose (mg/dL)	−1.06	1.42	0.45	−1.20	1.20	0.31	−1.10	1.21	0.36	1.37	2.18	0.53	1.20	1.67	0.47	2.52	7.57	0.74

* adjusted for mother’s age, daughter’s age, daughter’s BMI, and race. GEE, generalized estimating equation. BMI, body mass index. HDL, high-density lipoprotein. LDL, low-density lipoprotein.

**Table 4 ijerph-16-00015-t004:** Grandmaternal cardiovascular risk as a predictor of birth outcomes in grandchildren of female Bogalusa Heart Study participants.

Risk Factor		Birthweight (*n* = 110)								Gestational Age (*n* = 110)					
	unadjusted				adjusted				unadjusted				adjusted		
	beta	SE	*p*		beta	SE	*p*		beta	SE	*p*		beta	SE	*p*
BMI	−70	79	0.38		−12	86	0.89		−0.41	0.28	0.14		−0.49	0.30	0.11
subscapular skinfold	−67	76	0.38		1	73	0.99		−0.35	0.28	0.21		−0.33	0.23	0.15
weight gain during pregnancy with mother	20	59	0.74		−4	59	0.94		−0.15	0.23	0.51		−0.16	−0.12	0.23
systolic blood pressure (mmHg)	−34	48	0.48		2	52	0.97		0.27	0.20	0.18		0.34	0.22	0.13
diastolic blood pressure (mmHg)	9	44	0.84		51	42	0.23		0.06	0.13	0.67		0.17	0.15	0.26
HDL (mg/dL)	−56	46	0.23		−65	47	0.17		−0.30	0.17	0.08		−0.37	0.18	0.04
LDL (mg/dL)	−12	48	0.81		15	45	0.75		−0.01	0.16	0.93		0.07	0.14	0.62
total cholesterol (mg/dL)	−35	39	0.38		−55	41	0.17		−0.26	0.15	0.09		−0.40	0.17	0.02
triglycerides (mg/dL)	−66	49	0.17		−100	52	0.06		−0.11	0.21	0.61		−0.15	0.26	0.56
glucose (mg/dL)	−15	42	0.73		26	46	0.57		−0.02	0.16	0.91		0.03	0.18	0.86
	low birthweight in any pregnancy (*n* = 121, 17 cases)								preterm birth (*n* = 121, 13 cases)						
	unadjusted				adjusted				unadjusted				adjusted		
	odds ratio	95% CI			odds ratio	95% CI			odds ratio	95% CI			odds ratio	95% CI	
BMI	1.54	0.91, 2.58			1.40	0.80, 2.45			2.66	1.41, 5.02			2.59	1.35, 4.95	
subscapular skinfold	1.33	0.72, 2.48			1.09	0.53, 2.25			2.26	1.18, 4.35			2.07	1.13, 3.78	
weight gain during pregnancy with mother	0.78	0.44, 1.39			0.94	0.48, 1.81			0.95	0.52, 1.76			1.01	0.52, 1.93	
systolic blood pressure (mmHg)	1.34	0.79, 2.27			1.14	0.64, 2.05			0.65	0.40, 1.07			0.58	0.30, 1.12	
diastolic blood pressure (mmHg)	1.60	1.01, 2.54			1.44	0.82, 2.53			0.97	0.57, 1.67			0.80	0.42, 1.55	
HDL (mg/dL)	1.10	0.70, 1.74			1.12	0.70, 1.81			1.17	0.71, 1.91			1.18	0.72, 1.93	
LDL (mg/dL)	1.02	0.64, 1.62			0.90	0.53, 1.52			0.98	0.65, 1.5			1.04	0.70, 1.54	
total cholesterol (mg/dL)	1.05	0.69, 1.60			1.13	0.69, 1.84			1.10	0.69, 1.75			1.09	0.65, 1.82	
triglycerides (mg/dL)	1.07	0.67, 1.71			1.16	0.69, 1.94			0.97	0.49, 1.92			0.88	0.48, 1.59	
glucose (mg/dL)	1.36	0.90, 2.05			1.13	0.73, 1.74			1.30	0.82, 2.09			1.34	0.84, 2.15	

Adjusted for mother’s age, daughter’s age, daughter’s BMI, and race; GEE, generalized estimating equation. BMI, body mass index. HDL, high−density lipoprotein. LDL, low−density lipoprotein.

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
