# Peer review of "Maternal Pre-Pregnancy Cardiovascular Risk Factors and Offspring and Grandoffspring Health: Bogalusa Daughters"

_ijerph, 2018, doi:10.3390/ijerph16010015_

Round 1
Reviewer 1 Report
This study examined whether maternal obesity and cardiovascular risk factors are associated with daughter’s cardiovascular risk and grandchild birth outcomes — some concerns and statistical methods need to be addressed before considering for publication.
1. This study recruited 274 daughters and 208 mothers. Does it mean 2 or more daughters born to the same mother? How was this pair issue handled in the analysis? Also, whether daughters gave 2 or more births?
2. It is not clear when the mother’s adult measures were conducted – need to provide more details in the Method section. Also, whether the maternal pre-pregnancy measures were conducted separately for each daughter? For instance, if a mother had two daughters, there were two maternal pre-pregnancy measures.
3. Each analysis was based on different sample size, making it very difficult to compare the effect size between different risk factors.
4. In Table 4, it is not clear that the estimates for low birthweight and preterm birth were odds ratio or relative risk or beta?
5. It is needed to discuss further the role of maternal BMI and offspring BMI (i.e. the effect direction of lipids changed after adjusting for maternal BMI. What are the explanation and biological mechanisms?
6. Discussion: the authors concluded that there were some relationships between maternal adiposity, daughter’s cardiovascular risk factors, and grandchild birth outcomes, but in fact lack of power/effect to support this conclusion. I wonder whether this pilot study adds additional knowledge to current literature.
Author Response
1. This study recruited 274 daughters and 208 mothers. Does it mean 2 or more daughters born to the same mother? How was this pair issue handled in the analysis? Also, whether daughters gave 2 or more births?
Yes, this is the case. Generalized estimating equations were used to account for this clustering [page 4, line 141]. Some daughters did have two or more births; in that case, data from the first birth was used for analyses of birthweight and complications in any pregnancy were used for analyses of low birthweight/preterm birth [page 4, lines 148-150]. The number of pregnancies are also listed in Table 1.
2. It is not clear when the mother’s adult measures were conducted – need to provide more details in the Method section.
More information has been added to the methods.[page 3, lines 131-134] Given the study design, measures were not taken at a standardized time relative to the pregnancy, but mean and median age at this visit was 19 and 17 years old. This information was also included in Table 1. Median time between visit and daughter’s birth was 4 years.
Also, whether the maternal pre-pregnancy measures were conducted separately for each daughter? For instance, if a mother had two daughters, there were two maternal pre-pregnancy measures.
Yes, it was possible for these to be two different measures if there was more than one daughter in the sample.
3. Each analysis was based on different sample size, making it very difficult to compare the effect size between different risk factors.
Within tables, the sample sizes are not particularly inconsistent. Table 2 is based on approximately n≈170, table 3 on n≈170, and table 4 on n≈110. The exception is waist circumference in table 3, which was added later in the course of BHS and so has a smaller sample size. To make this easier to read, we have moved it to the end of the table, so the similarly-sized analyses are side-by-side.
4. In Table 4, it is not clear that the estimates for low birthweight and preterm birth were odds ratio or relative risk or beta?
The reviewer is correct; this should be an odds ratio. The error has been corrected.
5. It is needed to discuss further the role of maternal BMI and offspring BMI (i.e. the effect direction of lipids changed after adjusting for maternal BMI. What are the explanation and biological mechanisms?
A further discussion of BMI has been added to the discussion section [page 11, lines 234-245].
6. Discussion: the authors concluded that there were some relationships between maternal adiposity, daughter’s cardiovascular risk factors, and grandchild birth outcomes, but in fact lack of power/effect to support this conclusion. I wonder whether this pilot study adds additional knowledge to current literature.
We have reorganized the discussion to address specific topics more thoroughly and the pluses and minuses of this study [pages 11-12]. Given the relatively small number of studies of this design, their relatively small sample sizes, the fact that our results are generally consistent with previous work, and the lack of 3-generation studies in the US, we feel it makes a contribution to the scientific literature. In addition, this topic does not have a good translational model, so human studies are even more critical. Obese rodents (mice and rats), do not produce macrosomic offspring (the offspring of obese rodents are actually small for gestational age). Thus, infant birthweight is even more important to examine in these types of cohort studies in humans.

Reviewer 2 Report
The manuscript “Maternal pre-pregnancy cardiovascular risk factors and offspring and grandoffspring health: Bogalusa Daughters” reports data from a pilot study investigating the reproductive history of daughters from women enrolled in the Bogalusa Heart Study. Overall, some associations were found between maternal adiposity and cardiovascular risk, daughter’s cardiovascular risk and grandchild birth outcomes. There are a few concerns.
1. The introduction should be amended to include a true goal or hypothesis of the study.
2. Materials and methods: It is unclear what was done in this pilot study. In fact, there seems to be more methodological information in the abstract than in the methods section. This includes the participants for the current study and the measurements that were performed.
3. Table 1: Some sort of place marker should be included instead of blanks.
4. Tables 2-4: There are numerous columns which could be reorganized. It is hard to successfully consider all of this data. It is unclear what the headers are referring to, it is unclear what the values were adjusted for, etc.
5. In the Discussion section, the authors discuss the lack of many significant findings from the study and mention that some variables were different. Overall, there is a lack of specificity with which variables differed, were important, etc. This gives the impression that there were issues with the design, power, effect size, variables measured, etc.
Author Response
1. The introduction should be amended to include a true goal or hypothesis of the study.
A specific hypothesis has been added to the end of the introduction [page 2, lines 73-75].
2. Materials and methods: It is unclear what was done in this pilot study. In fact, there seems to be more methodological information in the abstract than in the methods section. This includes the participants for the current study and the measurements that were performed.
The methods section has been edited to follow the logic of the abstract more closely.
3. Table 1: Some sort of place marker should be included instead of blanks.
We defer to the editors in typesetting for journal style in terms of blank cells. We have added additional row markers to make the table easier to read and more consistent.
4. Tables 2-4: There are numerous columns which could be reorganized. It is hard to successfully consider all of this data. It is unclear what the headers are referring to, it is unclear what the values were adjusted for, etc.
Again, we defer to the editors for journal style for internal rules, etc. We have made the following changes to make the tables clearer:
Titles have been made more complete
Table 2: horizontal rules added to clarify pre-pregnancy vs. adult, sample size column removed since specified in the header; row and column headings made consistent
Table 3: order of outcomes rearranged as per reviewer 1; adjustment footnotes marked consistently; row headers made consistent with table 2
Table 4: row headers made consistent with table 2 and 3; columns aligned
Note that because of the large number of formatting changes, they are not tracked to avoid confusion
5. In the Discussion section, the authors discuss the lack of many significant findings from the study and mention that some variables were different. Overall, there is a lack of specificity with which variables differed, were important, etc. This gives the impression that there were issues with the design, power, effect size, variables measured, etc.
We now provide more details on the results in the discussion and more specificity about hypotheses and biological mechanisms.

Round 2
Reviewer 2 Report
Concerns were addressed
Author Response
Thank you for your kind comments.